# Cost Model Based Incremental Processing in Dynamic Graphs

**Kyoungsoo Bok [1], Jungkwon Cho [2], Hyeonbyeong Lee [2], Dojin Choi [2], Jongtae Lim [2] and Jaesoo Yoo [2,*]**

[1] Department of SW Convergence Technology, Wonkwang University, Iksandae 460, Iksan 54538, Korea; ksbok@wku.ac.kr

[2] Department of Information and Communication Engineering, Chungbuk National University, Chung-dae-ro 1, Seowon-Gu, Cheongju 28644, Korea; bruce1023@naver.com (J.C.); lhb@chungbuk.ac.kr (H.L.); mycdj91@chungbuk.ac.kr (D.C.); jtlim@chungbuk.ac.kr (J.L.)

\* Correspondence: yjs@chungbuk.ac.kr; Tel.: +82-43-261-3230

**Abstract:** Incremental graph processing has been developed to reduce unnecessary redundant calculations in dynamic graphs. In this paper, we propose an incremental dynamic graph-processing scheme using a cost model to selectively perform incremental processing or static processing. The cost model calculates the predicted values of the detection cost and processing cost of the recalculation region based on the past processing history. If there is a benefit of the cost model, incremental query processing is performed. Otherwise, static query processing is performed because the detection cost and processing cost increase due to the graph change. The proposed incremental scheme reduces the amount of computation by processing only the changed region through incremental processing. Further, it reduces the detection and disk I/O costs of the vertex, which are calculated by reusing the subgraphs from the previous results. The processing structure of the proposed scheme stores the data read from the cache and the adjacent vertices and then performs only memory mapping when processing these graph. It is demonstrated through various performance evaluations that the proposed scheme outperforms the existing schemes.

**Keywords:** dynamic graph; incremental processing; cost model; GAS model; cache

## 1. Introduction

A graph is a data structure for representing the multiple relationships between objects [1,2]. In recent real applications, dynamic graphs are generated, in which the vertices and edges constituting the graph are constantly changing [3–5]. These dynamic graphs are used to analyze changes in real time to create business value [6–8]. For example, the correlations between objects in the Internet of things (IoT) or disaster management are illustrated using dynamic graphs and analyzed in real time to detect and forecast disasters [9–11]. In a social network, an interaction change between items or users is modeled as a dynamic graph, and an event is detected or a recommendation service is provided [12–14].

Various large-scale graphs for processing have been conducted, as large volumes of dynamic graph are continually generated [15–20]. The distributed graph processing system was developed to store a large amount of graphs in a distributed manner and analyze the distributed graphs in parallel [21–24]. PowerGraph is a distributed processing system that stores subgraphs in a distributed manner across multiple servers and processes the graph in parallel to overcome the limitations of a single server system [16]. By adopting the vertex-cut technique and the gather-apply-scatter (GAS) model, data storage, and communications cost can be significantly reduced. Most of the graphs change continuously and increase in size gradually. However, most distributed graph processing systems perform static processing. If all the vertices and edges in a graph are processed each time a subgraph is changed, the processing cost is high, and real-time processing cannot be guaranteed. In other words, when a subgraph is changed, the static processing scheme calculates the entire graph, including the part that has not changed. Because this scheme performs unnecessary, i.e., redundant processing, it is extremely time consuming.

Owing to the problem of performing unnecessary, duplicate computations, the static processing schemes cannot provide real-time analysis results. Hence, an incremental processing scheme that processes only the changed part is required. To process continuous changes in a graph, the incremental processing scheme collects information on changes to the graph and processes only the part that has changed. Therefore, it solves the problem of redundant computations, which is the issue exhibited by the existing graph-processing scheme [25–27]. Because most of the graph analysis tasks have already been completed in the previous task, this scheme can perform more efficiently than the scheme that processes the entire graph when a small change has been implemented to the graph. In [28], unnecessary, redundant computations were removed utilizing an incremental processing scheme that calculated only the part that has changed in the graph. However, if many vertices were connected to the vertex that must be recalculated, numerous vertices must be detected to perform the recalculation, thus causing an issue of the disk I/O cost being proportional to the number of connected vertices. In [29], an incremental processing scheme that divides the unit of computation from the MapReduce [30] into smaller units and recalculates only the changed part was proposed. However, the unit of computation was divided into units that were extremely small. Therefore, the complexity was increased and resulted in a higher processing cost and network overhead. iGraph employed a method that divided the dynamic graph into several batch units and subsequently processed them [31]. However, partially redundant calculations occurred because the unchanged part, which was included in the batch, was recalculated when the graph was changed.

If the subgraph changed by the dynamic graph is large, the detection and processing costs may increase. If the entire graph is processed statically, only the processing cost is incurred. However, in the case of incremental processing, the region that must be recalculated owing to changes in the graph must be detected as well; hence, detection cost is incurred. Therefore, if the changed region is large, the method that processes the entire graph at once can outperform the incremental processing scheme that detects and subsequently processes the processing ratio.

In this paper, we propose an incremental dynamic graph-processing scheme called iGAS, which modifies the GAS model to perform incremental processing to provide real-time analysis results. The proposed scheme uses the cost model to selectively perform incremental processing or static processing based on the degree of change in the graph. The cost model calculates the predicted values of the detection and processing costs based on the actual processing history. The incremental processing is performed if it is more beneficial than static processing. The incremental processing uses the previous results through a cache strategy. The cache prefetches the read subgraph and adjacent vertices, and efficiently accelerates the repetitive computations.

This paper is organized as follows. Section 2 explains the related works. Section 3 describes the proposed scheme for incremental dynamic graph processing. Section 4 describes the results of the performance evaluation. Finally, Section 5 presents the conclusions of this study and the direction for future studies.

## 2. Related Works

A series of iterations are performed to process large-scale graphs. Each vertex receives the message sent from the previous iteration and forwards the message to another vertex. This type of vertex-based approach can process an extensive graph algorithm efficiently and accelerate the processing performance in a distributed processing environment. However, the graph algorithm can occasionally exhibit low memory access locality, and distributed processing across many servers worsens the locality and increases the chance of failure during calculation. Pregel proposed a bulk synchronous parallel model for processing large-scale graphs [15]. Pregel processed a sequence of iterations called superstep. During each superstep, a vertex can receive messages from the previous superstep in parallel, send messages to other vertices, and modify its own state and the state of its outgoing edges. Messages are typically sent along the outgoing edge, but messages can be sent to all known

vertices by the identifier. The processing terminates when no further vertices change in a superstep.

The limitations of a single server have been overcome by the development of distributed processing techniques to address steadily increasing data. However, data in the real world, such as social networks and the Web, exhibit a power-law graph distribution, in which several lower-level groups are connected to a single vertex. PowerGraph proposed the GAS model to solve the computation cost; storage cost incurred when processing graphs with the power-law distribution [16]. The GAS model divided the graph into subgraphs using the vertex-cut partitioning method and processed the subgraphs in parallel to effectively reduce the communication and processing costs. The GAS model utilized the intermediate message collection method as well as a technique to communicate between servers through shared memory. The GAS model processes graphs by repeatedly performing three phases: gather, apply, and scatter.

Graph analysis can produce meaningful results in various fields, such as social networking. Even though new, large-scale distributed processing systems have been proposed to accommodate the growing size of graphs, graph computation must be modified to accommodate specific graph algorithms or the application program family. It is difficult to build such systems because each distributed processing system comprises a different processing engine. GraphX [17] has been built on top of the Apache Spark [32], which is a widely used large-scale data processing engine for distributed graph processing and simplifies the implementation and application of algorithms. GraphX divides the vertices and edges into two tables and creates and partitions the resilient distributed datasets in each server using the vertex-cut method. Subsequently, GraphX executes the implemented graph algorithms on each server.

The MapReduce model, which was frequently used previously, has not been designed to process small changes efficiently. Hence, it does not support the incremental processing of large data. In [29], an incremental processing scheme that was compatible with the existing programs and did not require a separate new program was proposed. The scheme improved the MapReduce model to process small changes in the graph while maintaining the existing MapReduce model. To apply the incremental processing scheme to the MapReduce model, the computation unit must be divided into smaller units, and all calculations of the graph must be traced based on the modified propagation algorithm. Further, calculations are performed for only the changed part, and the results are propagated to the lower level such that they can be merged with the existing results for the part that has not changed.

The popularity of social networks has necessitated the real-time graph processing. However, most of the existing graph systems support the batch-processing method, Hence, dynamically processed graphs have incurred a high processing cost. iGraph [31] proposed a method that divided dynamic graph into several batches and subsequently processed them. When the graph was changed, iGraph employed a strategy to detect the batches that were affected by the change and recalculated the results that were then merged with the existing results. If the newly calculated values were less than the threshold value and hence deemed not affecting the adjacent vertices, the program would end. If the calculated values were greater than the threshold value, they were then propagated to the adjacent vertices.

Most of the graph algorithms are processed iteratively. However, the existing distributed processing systems only provided the batch processing method, resulting in the problem of recalculating the entire data when the graph was changed. In [28], an incremental processing scheme was proposed by dividing the graph by a region affected by the changes to the graph and a region unaffected by the changes. This scheme processed each region separately and merged the two results. Further, redundant calculations were avoided by incrementally processing the processing ratio of the graph and updating the results based on the changes.

The static graph processing is focused on quickly processing large volumes of data at once, while the incremental processing scheme is focused on reducing the amount of computation by calculating only the changed part. The static graph processing [15–17]

processes the entire graph; therefore, unnecessary, duplicate calculations occur, and real-time processing cannot be guaranteed [29]. Meanwhile, the incremental processing can result in a high processing cost and network overhead owing to increased complexity if the unit of computation is divided into units that are extremely small. In iGraph [31], partially redundant calculations occur because the unchanged part that is included in the batch is also recalculated when the graph is changed. Furthermore, if changes affect multiple batches, high disk I/O, and processing costs are incurred. In the incremental processing technique proposed by [28], if the vertex that must be recalculated involves many connections, and the connected vertices must be detected again. Therefore, the detection and disk I/O costs proportional to the number of connected vertices are incurred. The incremental processing must be adopted to provide real-time analysis results and to avoid the problem of performing redundant computations in static processing. To solve the issue of partially redundant computations inherent in the existing incremental processing techniques, an efficient incremental processing scheme that reduces the detection and disk I/O costs, as well as a cache strategy, are required. Furthermore, an efficient, differentiated graph-processing technique is required based on graph changes.

### 3. The Proposed Incremental Graph-Processing Scheme

*3.1. Overall Procedure*

When a graph is changed, it may be more efficient to calculate only the changed part rather than computing the entire graph. The existing incremental processing schemes encounter the problem of having to again detect the adjacent vertices connected to the vertex that is being recalculated. To reduce the processing cost incurred by detecting the adjacent vertices in addition to the disk I/O cost, we propose a technique to reuse the previous results as well as a cache strategy to accelerate incremental processing. In addition, because the processing costs to detect and process the graph in the processing ratio increase according to the changes, a cost model that selectively performs static processing or incremental processing is proposed. If incremental processing is more beneficial than static processing, the cost model detects the region affected by the changes in the graph and performs incremental processing in the affected region to generate results.

Figure 1 shows the overall procedure of the proposed scheme. The proposed scheme selectively performs both static processing and incremental processing based on the cost model when graph is changed. To determine a query-processing method, first, query processing statistics are calculated based on past query-processing history. When the graph is changed, the recalculation cost is predicted using query-processing statistics. The processing method decision calculates a cost model using the recalculation cost prediction. If there is a benefit in the cost model, the proposed scheme, known as incremental gather-apply-scatter (iGAS), is performed to incrementally process the region affected by graph changes. If incremental processing is not beneficial, then GAS processes the entire data statically. When iGAS is performed, the cache strategy is used. Therefore, subgraphs that are read once and the calculated results are stored. The subgraphs from adjacent vertices are prefetched to enhance the efficiency of incremental processing and reduce the disk I/O cost.

*3.2. Recalculation Cost Prediction*

The proposed scheme uses a cost model to selectively perform static processing and incremental processing. In order to select a query-processing method, the cost model must predict the cost of static processing and incremental processing costs according to graph changes. Because static processing reads and processes the entire graph, it is affected by the size of the graph and the cost of processing vertices. On the other hand, the incremental processing determines the recalculation area according to the graph change, and requires a processing cost for the recalculation area. The recalculation cost prediction calculates the statistical values, such as the average number of vertices that must be recalculated ($NRV$),

the statistics of the detection cost (*SDC*), and the statistics of the processing cost (*SPC*) that affect the query-processing cost through past query-processing history.

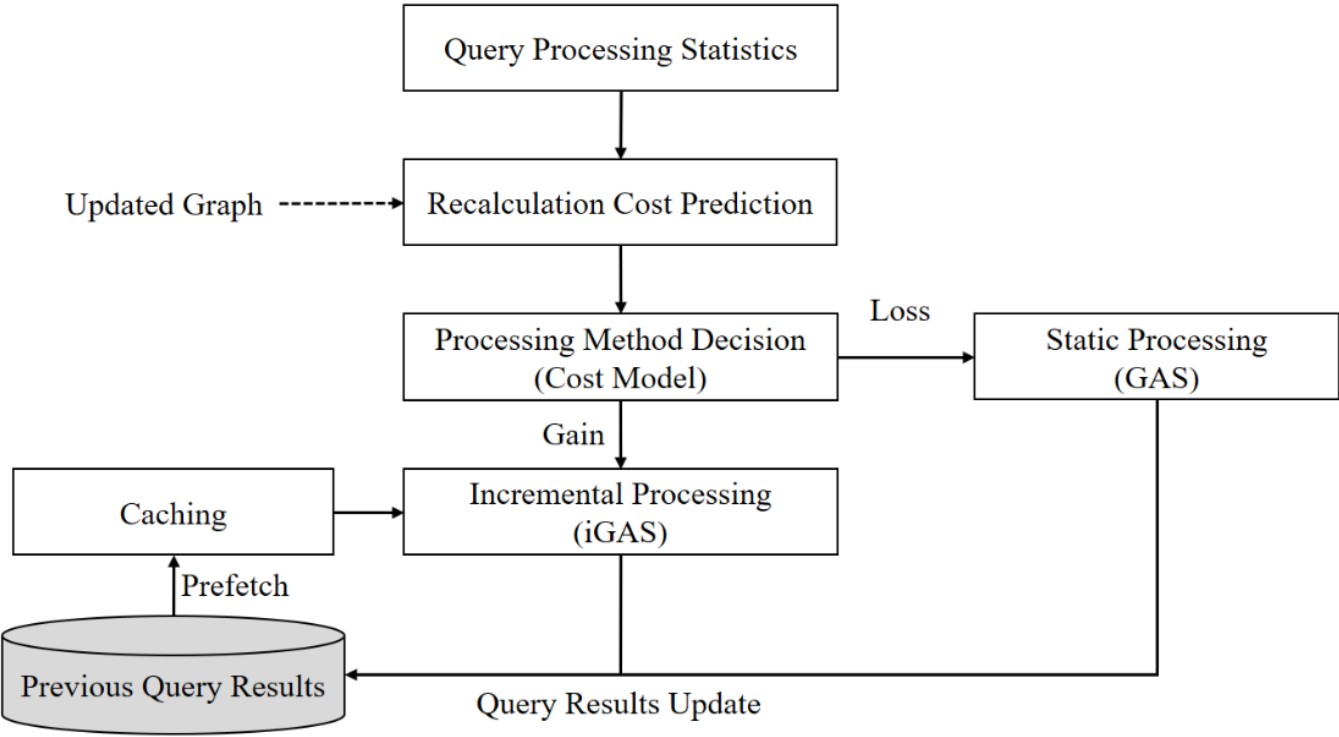

**Figure 1.** Overall procedure of the proposed scheme.

*NRV*, *SDC*, and *SPC* are calculated by Equations (1)–(3), where $NU_i$ is the number of change operations contained in *i*-th graph, $NAV_i$ is the number of vertices affected by the graph change, $DC_i$ is the detection cost of the actual processing ratio, and $APC_i$ is the actual processing cost predicted using past statistics. *NRV* is the average number of vertices that must be recalculated because they were affected when a single vertex was changed. The number of affected vertices increases if the vertex is connected to many vertices. Hence, the number of vertices that must be recalculated increases proportionally. *NRV* is calculated by collecting the number of vertices affected by the graph change based on the number of changes in the dynamic graph and the actual processing history of iGAS *SDC* refers to the cost incurred when detecting a single vertex. If the amount of the change is large, the detection cost increases and the past statistics of the detection cost increases proportionally. The past statistics of the detection cost is calculated by collecting the number of vertices affected by the graph change and the detection cost of the actual processing ratio. *SPC* is the cost of processing a single vertex, and this value can change fluidly according to the system load. The past statistics of the processing cost is calculated by collecting the number of vertices affected by the graph change and the actual processing cost.

$$NRV = \frac{\sum_{i=1}^{n} \frac{NAV_i}{NU_i}}{n} \tag{1}$$

$$SDC = \frac{\sum_{i=1}^{n} \frac{DC_i}{NAV_i}}{n} \tag{2}$$

$$SPC = \frac{\sum_{i=1}^{n} \frac{APC_i}{NAV_i}}{n} \tag{3}$$

*3.3. Cost Model*

In dynamic graphs, there are query types that must continuously provide query results whenever the vertices and edges in the graph change. For example, a continuous query that executes the same query during a specific time re-executes the query processing whenever a graph change occurs to change the query result. In an application that does not significantly affect the query result due to partial addition or deletion of vertices and edges, if only the changed part is processed through incremental processing, the number of disk I/O, and the overall processing time can be reduced. However, as incremental processing includes the cost of detecting the recalculation region and the cost of query processing for the recalculation region, it may be effective to statically process the entire graph if there are many subgraphs affected by the graph change. The cost model selectively performs either the incremental processing scheme or static processing scheme based on the query-processing cost according to graph changes to improve the processing efficiency.

The incremental processing cost ($CIP$) is determined by the detection cost ($DC$) for the recalculation region and the processing cost ($PC$) for the recalculation region, as shown in Equation (4). $DC$ is calculated by Equation (5), where $NRV$ and $SDC$ are the average number of vertices that must be recalculated and the statistics of the detection cost determined in recalculation cost prediction, and $NU$ is the number of changes. $PC$ is calculated by $NRV$, $SPC$, and $NU$ as shown in Equation (6), where $SPC$ is the statistics of the processing cost. In the incremental processing cost, the detection cost and processing cost increase because the number of vertices affected by graph change increases when $NRV$ and $NU$ are large. In addition, as the $SDC$ and $SPC$ increase, it takes a lot of time to determine the recalculation region. As a result, the processing time increases.

$$CIP = DC + PC \qquad (4)$$

$$DC = NRV{\cdot}SDC{\cdot}NU \qquad (5)$$

$$PC = NRV{\cdot}SPC{\cdot}NU \qquad (6)$$

The static processing cost ($CSP$) is determined by the number of vertices ($NG$) and $SPC$, as shown Equation (7). As the static processing cost processes the entire graph, the processing cost increases as the number of vertices included in the entire graph increase.

$$CSP = NG{\cdot}SPC \qquad (7)$$

The cost model ($CM$) is the difference between the predicted cost of static processing and that of incremental processing, and it is expressed by Equation (8). If $CM$ is negative, it means that it takes a lot of time to determine the recalculation region according to the graph change, and the number of vertices included in the recalculation region increases, which means that the cost of incremental processing increases compared to static processing. When $CM$ is negative, it is more effective to process the entire graph rather than incremental processing. If $CM$ is positive, it is more effective to perform incremental processing because the cost of processing the vertices included in the entire graph is higher than determining and processing the recalculation area according to graph changes. The incremental processing performs better than static processing when the number of vertices affected by the change is reduced.

$$CM = CSP - CIP \qquad (8)$$

*3.4. Incremental Processing*

If the result of the cost model demonstrates that the incremental processing scheme is not beneficial because the graph has changed significantly, static processing is performed using the existing GAS. If the amount of change in the graph is small and the incremental processing technique is deemed to be beneficial, then the cost model performs incremental processing utilizing iGAS. This paper proposes an iGAS model that was derived by

modifying the GAS model to perform incremental processing such that real-time analysis results could be obtained. The GAS model was proposed by PowerGraph [16], and it is a distributed graph processing system. To overcome the limitations of a single server, the GAS model stores data across multiple servers and performs distributed processing. The proposed iGAS model processes only the changed part. Therefore, it does not perform redundant calculations. Further, the cost for redetecting the data is reduced because the previously generated result data are reused.

PowerGraph processes data using the static processing scheme. Therefore, the entire region (a) of Figure 2 is calculated. Hence, even the unaffected regions are recalculated. For the existing incremental processing technique, if vertex $v_2$ is inserted into region (a) of Figure 2, vertex $v_1$, which is directly affected by vertex $v_2$, and vertices $v_3 \sim v_5$, which are subsequently affected by vertex $v_1$ are included in the processing ratio. However, vertices $v_6 \sim v_{10}$ must be included in the processing ratio to calculate vertex $v_1$. If many connections exist for vertex $v_1$ when it is to be calculated, as in region (b), the number of times that data are redetected and calculated in the existing incremental processing scheme is equivalent to the number of vertices connected to vertex $v_1$. To improve this problem, the GAS model is expanded to reuse the previously generated result data to reduce the redetection cost. Through this enhancement, only the region that is affected by the graph change—region (c) in this case—is processed and the data processing time is reduced.

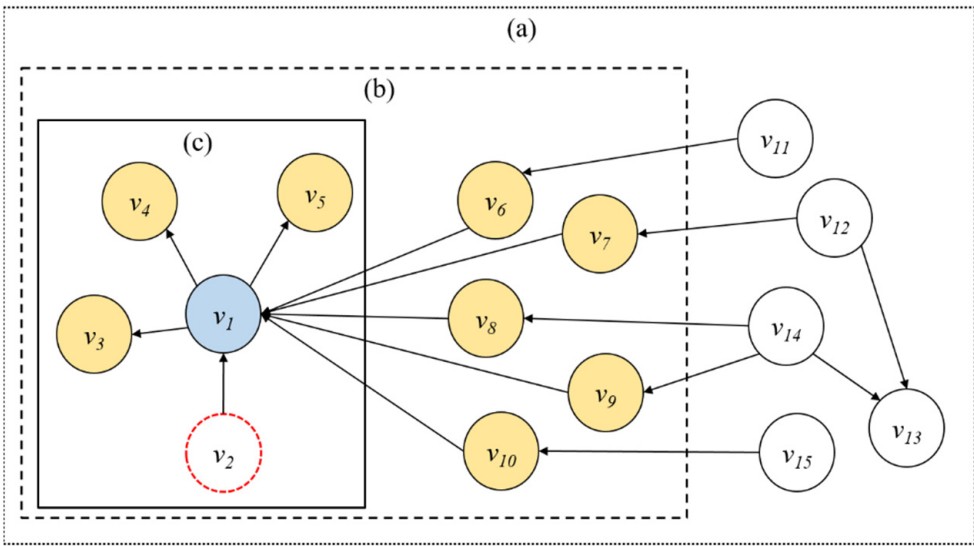

**Figure 2.** Processing ratio.

The GAS model is a computational model designed to process graphs and is divided into three phases: gather, apply, and scatter. Similar to Pregel, the gather phase receives messages by intermediately gathering information processed by servers in a distributed environment and collects information from the connected vertices. In the apply phase, the collected information is computed. Further, the calculated results are shared between the servers through shared memory. In the scatter phase, the calculated results are propagated to the neighboring vertices. The GAS model performs these three phases iteratively.

Algorithm 1 shows the existing GAS model. Complex and iterative types of graph operations, such as those of most data mining and machine-learning algorithms, request information from neighboring vertices iteratively to compute the value of the target vertex. Therefore, the GAS model is efficient from the structural aspect of the graph analysis algorithms. Further, for the complex algorithm that uses adjacent vertices to perform computations, PowerGraph has generated the analysis results the fastest among the existing distributed graph-processing systems. Therefore, [33,34] used the GAS model to perform static processing.

---

**Algorithm 1.** GAS Vertex-Program

---

**Input**: center vertex $u$
**foreach** neighbor $v$ is empty
   $a_u \leftarrow$ sum($a_u$, gather($D_u$, $D_{(u,v)}$, $D_v$))
**end**
$D_u{}^{new} \leftarrow$ apply($D_u$, $a_u$)
**foreach** neighbor $v$ scatter_nbrs($u$)
   ($D_{(u,v)}$, $\triangle a$) $\leftarrow$ scatter($D_u$, $D_{(u,v)}$, $D_v$)
**end**

---

The proposed scheme modifies the existing GAS model to support incremental processing such that it only recalculates the changed region. Algorithm 2 is the proposed iGAS model. Among the adjacent vertices, the previous and present values of the vertex being changed are collected and stored in the cache during the gather phase of the iGAS model. If the previous value exists in the cache owing to the cache policy, the value is collected from the cache. If the value does not exist in the cache, it is read from the disk and then stored in the cache. Once the gather phase has been completed, a new value of the vertex is calculated using the data of the changed vertex in the apply phase. For the PageRank [35] algorithm, an inverse function is used to calculate the new value by modifying only the vertex value of the changed part among the values collected from the remaining adjacent vertices. In the scatter phase, because the vertex thereof that has been changed is recorded in the adjacent vertices, the newly calculated value is propagated.

---

**Algorithm 2.** iGAS Vertex-Program

---

**Input**: center vertex $u$
**foreach** neighbor changed vertex $c_v$ in gather_nbrs($u$)
   **if** ($c_{v, flag}$ == 1) **then**
      $c_u \leftarrow$ fix($c_u$, gather($c_{v.id}$, $c_{v.old\_val}$, $c_{v.new\_val}$, $c_{v.deg}$))
   **end**
**end**
$D_u{}^{new} \leftarrow$ apply($D_u$, $c_u$)
**foreach** neighbor $v$ scatter_nbrs($u$)
   delta = ($D_{(u,v)}$, $\triangle a$)
   **if** delta is not converged **then**
      $D_{u.flag}$ = 1
      delta $\leftarrow$ scatter($D_u$)
   **end**
**end**

---

GAS iterates the gather, apply, and scatter phases until there are no active vertices remaining participating in the user-defined query. When the vertex program processing graphs that are centered around the vertices complete the scatter phase, it becomes inactive until it is reactivated. One of the methods for improving graph-processing performance is to reduce the number of vertices participating in the calculation. Therefore, the complexity is expressed as a computation cost according to the number of vertices participating in graph processing. When a query that performs $k$ iterations is processed, Equation (9) is the complexity of GAS *(CG)*, where $CG_i$, $CA_i$, $CS_i$ are the average processing costs when each vertex process the gather, apply, and scatter phases in the $i$-th iteration. If $SV$ is the set of vertices and $IV_i$ is the set of inactive vertices in the $i$-th iteration, then $n(SV - M_i)$ is the number of active vertices when processing the $i$-th iteration in GAS. $IV_1$ is zero because all vertices included in $SV$ are activated in the first iteration. Therefore, $CG$ can be expressed as Equation (10). Finally, $CG$ is Equation (11) because $n(SV) \cdot (CG_1 + CA_1 + CS_1)$ is $CSP$ by Equation (7). Equation (12) is the complexity of iGAS $(CIG)$, where $DC$ is the detection cost calculated by Equation (4), $RV$ is the set of vertices included in the recalculation region

and $n(RV - IV_i)$ is the number of active vertices when processing the $i$-th iteration in iGAS. $IV_1$ is zero because all vertices included in $RV$ are activated in the first iteration. Therefore, $CIG$ can be expressed as Equation (13). As $n(RV) \cdot (CG_1 + CA_1 + CS_1)$ is $PC$ calculated by Equation (6), $CIG$ is expressed as Equation (14). Finally, $CIG$ is Equation (15) because $CIP$ is $DC + PC$ by Equation (4). iGAS is selected when $CSP$ is greater than $CIP$. If $i \geq 2$, $n(SV - IV_i)$ is greater than $(RV - IV_i)$. As a result, the complexity of iGAS is smaller than that of GAS.

$$CG = \sum_{i=1}^{k} n(SV - IV_i) \cdot (CG_i + CA_i + CS_i) \tag{9}$$

$$= n(SV) \cdot (CG_1 + CA_1 + CS_1) + \sum_{i=2}^{k} n(SV - IV_i) \cdot (CG_i + CA_i + CS_i) \tag{10}$$

$$= CSP + \sum_{i=2}^{k} n(SV - IV_i) \cdot (CG_i + CA_i + CS_i) \tag{11}$$

$$CIG = DC + \sum_{i=1}^{k} n(RV - IV_i) \cdot (CG_i + CA_i + CS_i) \tag{12}$$

$$= DC + n(RV) \cdot (CG_1 + CA_1 + CS_1) + \sum_{i=2}^{k} n(RV - IV_i) \cdot (CG_i + CA_i + CS_i) \tag{13}$$

$$= DC + PC + \sum_{i=2}^{k} n(RV - IV_i) \cdot (CG_i + CA_i + CS_i) \tag{14}$$

$$= CIP + \sum_{i=2}^{k} n(RV - IV_i) \cdot (CG_i + CA_i + CS_i) \tag{15}$$

If the graph is changed, iGAS detects and calculates the part affected by the graph change, as shown in Figure 3. Most of the graph algorithms perform many comparison operations. Therefore, a large overhead will occur in the cost of detecting the processing ratio if the algorithms are implemented using a linked list or array. Thus, the cost of detecting the processing ratio should be reduced by implementing the algorithms using a hash map to access the data.

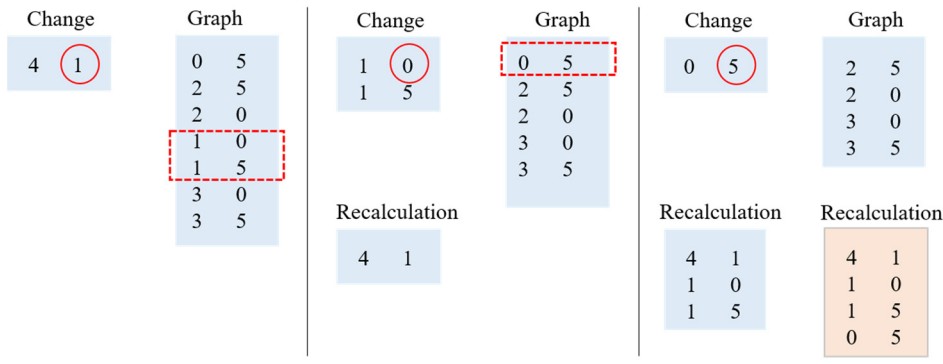

**Figure 3.** Processing ratio detection in iGAS.

Figure 4 shows how the proposed scheme reuses the data from the previous results iteratively. The proposed scheme only collects the previous result data of the changed vertex and does not need to collect the entire data that are connected to the vertex. Therefore, the proposed scheme can reduce the disk I/O cost more than the existing incremental processing scheme.

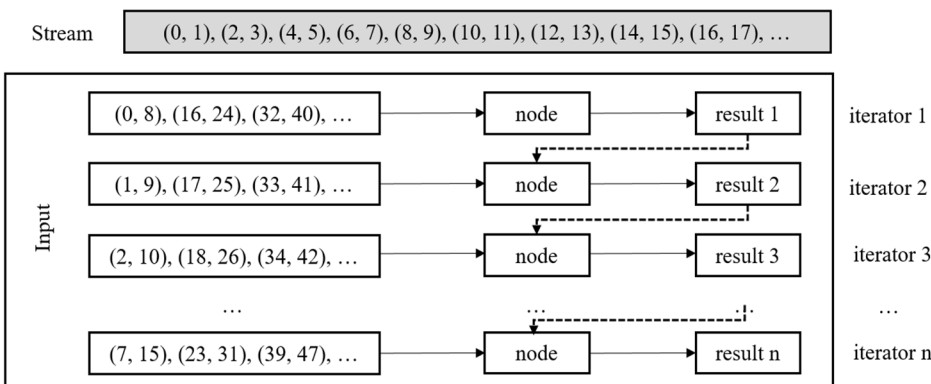

**Figure 4.** Reuse of previous results in iGAS.

Figure 5 shows the vertices that are collected while iGAS is performed. If the graph change changes vertex $v_1$, the vertex being calculated—vertex $u$—gathers the previous result data and the present result data of the changed vertex $v_1$ among the adjacent vertices, and the new value of vertex $u$ is calculated in the apply phase. The newly calculated value of vertex $u$ is propagated to the adjacent vertex $v_4$, and the new value of vertex $v_4$ is computed by only collecting the value of vertex $u$. When the dynamic graph is processed using the proposed incremental processing scheme, the duplicate processing of the data and the cost of redetecting the connected vertices are reduced, thus in turn enhance the processing time.

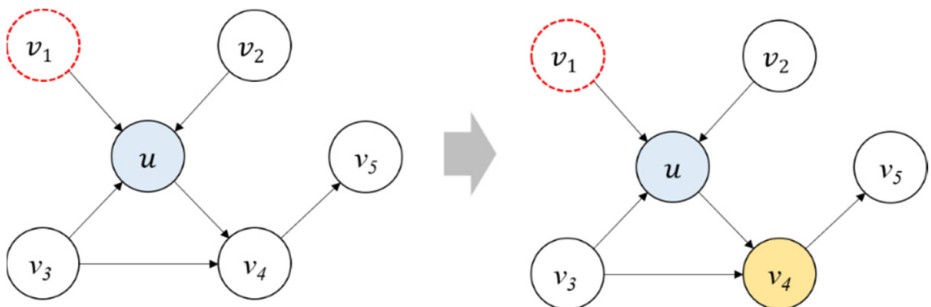

**Figure 5.** Vertices collected during incremental processing.

*3.5. Caching*

To perform the incremental processing scheme efficiently, the data processing results must be reused. When data are processed, it is highly likely that the data once used will be used again. Furthermore, the data locality is high, such that the probability of using the surrounding data is high as well. Therefore, storing such data in the cache can enhance the processing performance. In this study, the cache strategy is used when iGAS is performed; therefore, data once read and the calculated result data are stored, and the data of adjacent vertices are prefetched. Utilizing the cache strategy improves the efficiency of incremental processing and reduces the disk I/O cost. If the cache is full, the LRU (least recently used) technique [36] is used to manage the cache. During the gather phase, if the previous and current values of the data being changed are not in the cache, they are then stored in the cache. In the subsequent operations, the cached data are used when the value of this vertex is read such that the data processing time can be enhanced.

Apart from the once read data, the data of adjacent vertices are stored in the cache in advance while the iGAS model is processing the vertex. Through this mechanism, only memory mapping is performed when the graph is processed, and the efficiency is improved. Figure 6 shows the cache management policy. When the dynamic graph are entered, the hash key value of the data is created using the hash map. The hash key value is obtained by computing the hash function using the number of vertices and information

such as the change status, previous value, present value, and number of out-degree edges. Subsequently, iGAS is performed and when the vertex matching the hash key value in the cache is calculated, the data of the adjacent vertex that have changed are collected and used in the processing. The new results processed by iGAS are stored in the cache, and subsequently fetched from the cache and reused when the adjacent vertex is calculated.

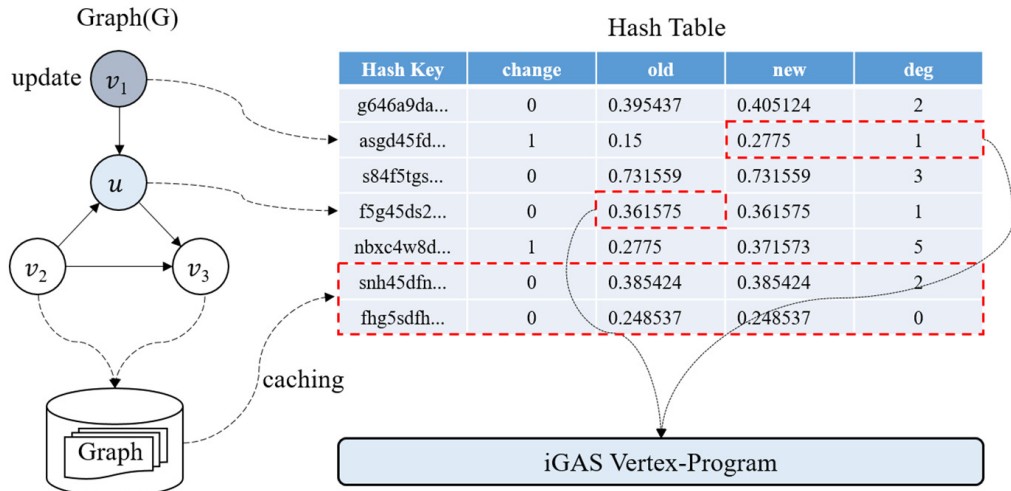

**Figure 6.** Cache management.

## 4. Performance Evaluation

The performances of the static processing technique that computes the entire graph and the incremental processing scheme that calculates only the changed part are compared to demonstrate the superiority of the proposed scheme. The processing times of PowerGraph [16], which is the representative static processing technique, the scheme of [28] that supports incremental processing, and the proposed technique, are compared in the performance evaluation. The performance evaluation environment consisted of the Intel® Core i5 CPU processor, 32GB memory, and 1TB disk. Table 1 shows the characteristics of the experimental data. In the experiment, Live Journal [37], Twitter [38], and Google Web [39] provided by the Stanford Large Network Dataset collection [40] were used to compare the performance of the cost model as well as the processing time.

**Table 1.** Characteristics of the experimental dataset.

| Dataset | Vertices | Edges |
|---|---|---|
| Live Journal | 1,070,383 | 3,372,093 |
| Twitter | 81,543 | 2,419,738 |
| Google Web | 875,713 | 5,105,039 |

To prove the validity of the cost model, Figure 7 shows the processing times using the only iGAS and the scheme that utilizes both the iGAS and cost models in comparison with the static processing scheme. It takes longer to execute iGAS than the static processing scheme when the processing ratio is over 55%. If most of the graph has to be recalculated owing to the graph change, the cost to detect the processing ratio is included in iGAS. Hence, iGAS is not more beneficial than static processing in this case. The performance evaluation was conducted after the cost model was applied to iGAS such that selective processing can be performed based on the changes in the graph. If incremental processing is inefficient based on the cost model, then the static processing scheme is used to process the entire data without detecting the changed part. Moreover, the processing times were added up for cases from 5% processing ratio to 100% processing ratio for each scheme and subsequently compared. When only iGAS was used, the sum of the processing times

was about 97%. Therefore, it was more time consuming than the static processing scheme. When the iGAS and cost models were used together, the sum of the processing times was about 141%. Hence, the results verify that it is more advantageous to utilize the cost model than to use iGAS only.

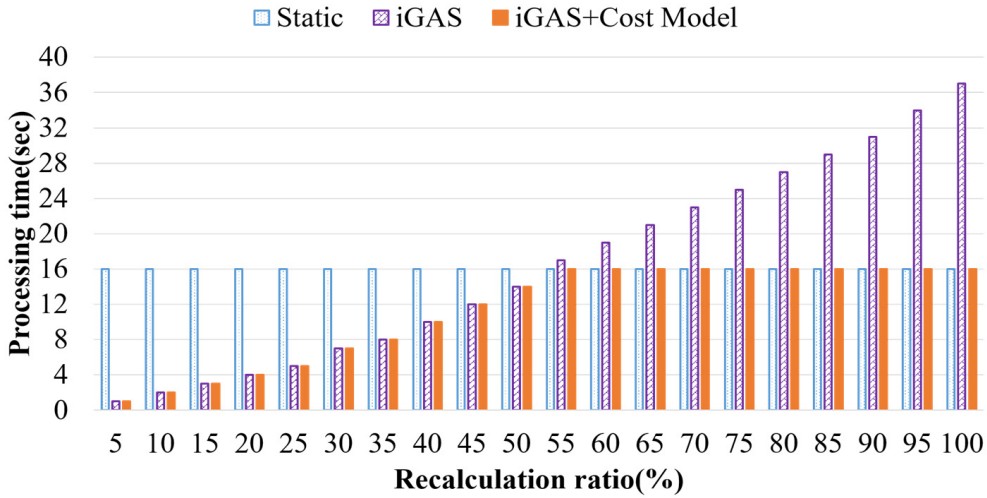

**Figure 7.** Comparison of processing times based on recalculation ratio.

If a change is performed to a particular vertex that contains many connections, the cost for detecting the processing ratio may increase for the proposed cost model. Further, an error may occur because the detection cost is computed based on the actual processing history of the cost model. To verify whether a correction choice has been selected based on the predicted costs of static processing and incremental processing calculated by the cost model, a performance evaluation was conducted. Decision-making capability (*DMC*) is used to determine whether the correct selection has been made by using Equation (16), where *AC* is the actual incurred cost and *PC* is the predicted cost.

$$DMC = 1 - \left| \frac{AC - PC}{AC} \right| \tag{16}$$

To determine the decision-making capabilities for the predicted costs of static processing and incremental processing, the PageRank algorithm was performed 25 times for each data type. Figure 8 shows the decision-making capabilities for the predicted costs of static processing and incremental processing. In Figure 8a, the decision-making capability for the predicted cost of static processing is 95% on average, which is excellent. However, the decision-making capability for the predicted cost of incremental processing in Figure 8b does not exceed 50% between the first and 16th iterations. Between the 17th and 25th iterations, the average decision-making capability is 75%. The decision-making capability for the predicted cost of static processing is excellent; therefore, we know that the correct selection has been made. However, if the decision-making capability is low in the early iterations as with the predicted cost of incremental processing, then a type-1 error occurs and could result in the problem of performing static processing when the incremental processing technique is advantageous. This result occurred because the cost model computes the predicted values based on the actual processing history, and the processing was performed only a small number of times. Further, the analysis results indicated that the decision-making capability was significantly reduced when calculating a vertex with many connections.

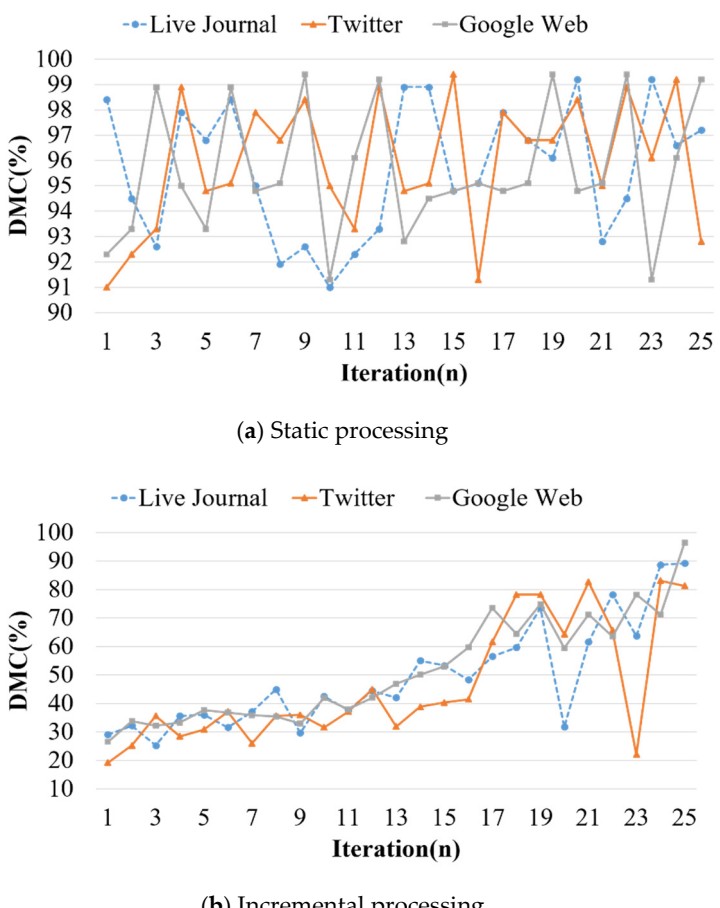

(**a**) Static processing

(**b**) Incremental processing

**Figure 8.** Decision-making capability of static processing and incremental processing.

To compare the performances of the existing schemes [16,28], the proposed iGAS scheme, and the iGAS scheme that incorporated the cost model, the PageRank algorithm and the single source shortest path (SSSP) algorithm [41] were performed 10 times for each data set. The results for the 10 iterations were added up. Figure 9 shows the processing time of the proposed schemes and the existing schemes on three types of data sets. The performance evaluation was conducted by updating the graph for each iteration. For cases where the graph was changed, the static processing scheme [16] was inefficient because the entire data were calculated, thus producing duplicate results. The existing incremental processing [28] computed only the part affected by the graph change. However, the incurred detection and disk I/O costs were proportional to the number of connections of the vertex. Hence, it was verified that the processing time of this technique was longer than that of the iGAS technique. When compared with the existing incremental processing scheme for performing the PageRank algorithm, the iGAS scheme was 146% superior in performance on average, and the scheme that used both the iGAS and cost models was 198% superior in performance on average. When compared with the existing incremental processing scheme for performing the SSSP algorithm, the iGAS scheme was 161% superior in performance on average, and the scheme that used both the iGAS and cost models was 226% superior in performance on average. When only the iGAS technique is applied, the processing cost may exhibit a performance that is similar to that of the static processing cost if most of the graphs have been changed. However, the performance of this technique may be worsened because an additional detection cost is incurred in proportion to the amount of change. By utilizing both incremental processing and the cost model, the proposed scheme demonstrated excellent performance.

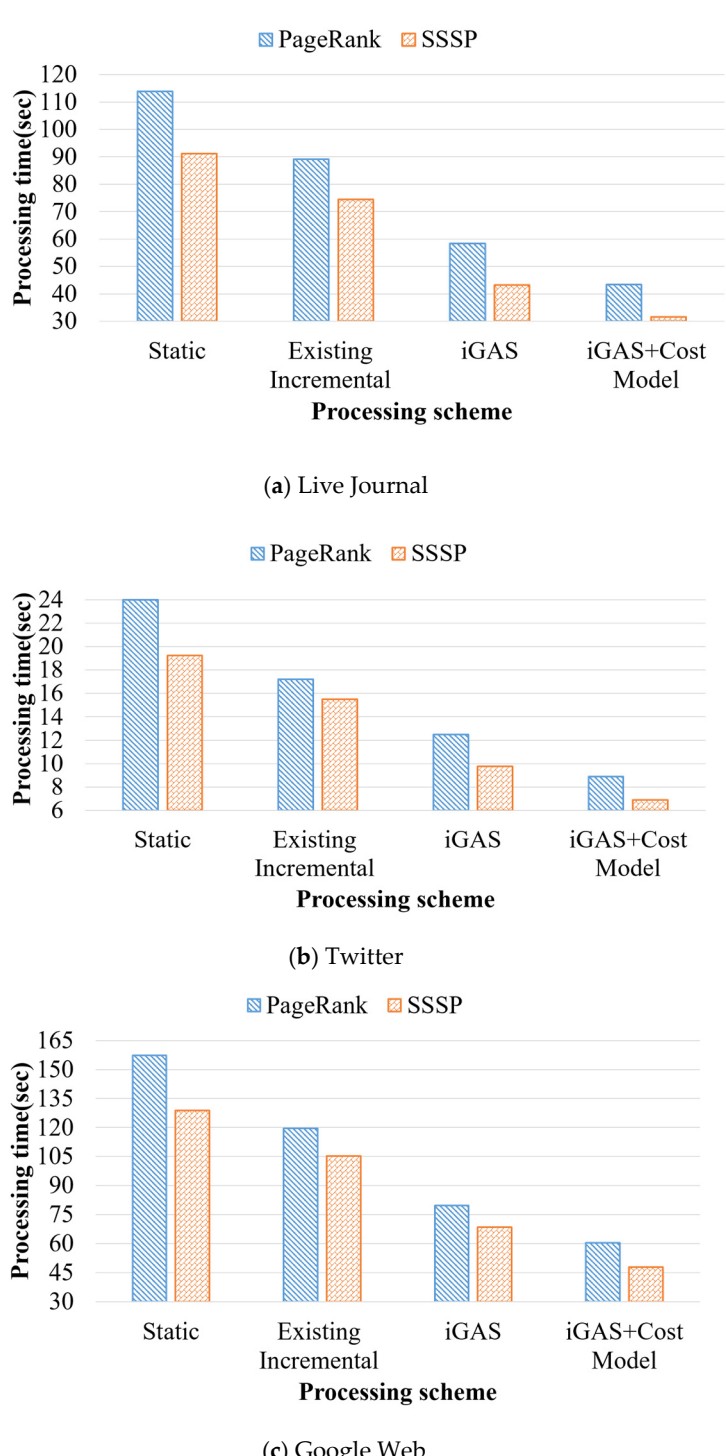

(**a**) Live Journal

(**b**) Twitter

(**c**) Google Web

**Figure 9.** Processing times on three types of data sets.

## 5. Conclusions

In this paper, we proposed an efficient scheme that processes dynamic graph incrementally by data reuse. When a vertex was connected to many vertices, the proposed scheme reused the previously generated result data. Hence, the detection and disk I/O costs of the connected vertices were reduced. Further, the cost model predicted the processing cost, and either the static processing technique or incremental processing technique was selectively performed based on the processing ratio. When compared with the existing incremental scheme, the performance evaluation results verified that the processing time improved by 212% on average for the proposed scheme. For future studies, a method to

improve the decision-making capability of a particular vertex that has many connections will be investigated.

**Author Contributions:** Conceptualization, K.B., J.C., H.L., D.C., J.L. and J.Y.; methodology, K.B., H.L., D.C., J.L. and J.Y.; software, J.C., D.C. and J.L.; validation, K.B., J.C. and D.C.; formal analysis, K.B., J.C. and J.Y.; data curation, J.C.; writing—original draft preparation, K.B., J.C. and J.Y.; writing—review and editing, K.B., J.C. and J.Y. All authors have read and agreed to the published version of the manuscript.

**Funding:** This work was supported by the National Research Foundation of Korea(NRF) grant funded by the Korea government(MSIT) (No. NRF-2020R1F1A1075529), by Institute of Information & communications Technology Planning & Evaluation (IITP) grant funded by the Korea government(MSIT) (No.2021-0-02082, CDM_Cloud: Multi-Cloud Data Protection and Management Platform), under the Grand Information Technology Research Center support program(IITP-2022-2020-0-01462) supervised by the IITP(Institute for Information & communications Technology Planning & Evaluation), and was carried out with the support of "Cooperative Research Program for Agriculture Science and Technology Development (Project No. PJ01624701)" Rural Development Administration, Republic of Korea.

**Conflicts of Interest:** The authors declare no conflict of interest.

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
