# Peer review of "Cost Model Based Incremental Processing in Dynamic Graphs"

_electronics, doi:10.3390/electronics11040660_

Round 1

Reviewer 1 Report

The authors propose a new and nice incremantal dynamic graph processing scheme by means of a cost model for being able to selcetively performing two processing methods, incremental processing or static processing. There are cases where each method is proven to be more useful and practical. So the proposed scheme will be quite attractive to the researchers in the area. The proposed scheme has the advantage over the existing ones of being reducing the amount of computational work and also processing cost of recalculations based on the past processing history after each  update of the vertices and edges of the graph. The data processing system for dynamic graphs called iGraph introduced in 2016 is succesfully implemented in this manuscript. 

The methods in the results are clearly stated together with their advantages over the existing equivalents. There is a satisfactory number of figures explaining the decision making, processing schemes, processing times, etc.

It is an interesting work and should be widely used by others. There are a few grammatical mistakes which I overlined in the submitted pdf file. Authors should go through the manuscript and check all of it for similar or the same mistakes.

Afte these corrections are done, it is acceptable.

Author Response

Dear Reviewer,

We would like to sincerely thank you for your attentive indications and good comments. We tried to do our best in order to revise and complement your comments. Please refer to the attached file on the detailed revisions.

Many thanks.

Jaesoo Yoo

Reviewer 2 Report

This paper developed a strategy to accelerate the incremental processing used for solving the redundant computation in distributed graph processing, and presented a cost model to selectively perform the static processing or incremental processing. 

  1. The contribution of the paper is somewhat limited owing to three aspects. 1) In sec. 3.2, only several notions are defined, the usage of the recalculation cost prediction is not justified. 2) The authors claim that the incremental processing technique is deemed to be beneficial if the amount of change is small, it is a very rough claim without sufficient theoretical analysis.  3) The cost for binary selection of using static processing or incremental processing is not clarified.
  2. Which kinds of graphs are suitable for running the incremental processing? The authors should clarify.
  3. The complexity of Algorithm 1 and  Algorithm 2 should be compared. The complexity of the overall algorithm should be theoretically analyzed, only the processing time (see Fig7) is not sufficient.

Minors:

  1. Line 100, what is meant by ‘the its’?
  2. The title of sec. 3.3 seems incorrect.
  3. The caption of Fig.9 and the content of Figure are separated into two different pages.

Author Response

(The authors gave the same response as above.)

Round 2

Reviewer 2 Report

Most of previous comments have been addressed, except that the computational complexity analysis in section 3.4 seems not explicit. In general, the complexity is measured by $O( )$.

Author Response

Dear reviewer,

We would like to sincerely thank you for your attentive indications and good comments. Our paper was partially rewritten in order to revise and complement your comments. Please refer to the attached file about the detailed revisions.

Many thanks.

Jaesoo Yoo
